# The Combined Immunohistochemical Expression of GLI1 and BCOR in Synovial Sarcomas for the Identification of Three Risk Groups and Their Prognostic Outcomes: A Study of 52 Patients

**DOI:** 10.3390/ijms25147615

**Published:** 2024-07-11

**Authors:** Francisco Giner, Emilio Medina-Ceballos, Raquel López-Reig, Isidro Machado, José Antonio López-Guerrero, Samuel Navarro, Luis Alberto Rubio-Martínez, Mónica Espino, Empar Mayordomo-Aranda, Antonio Llombart-Bosch

**Affiliations:** 1Pathology Department, Hospital Universitari i Politècnic La Fe of Valencia, 46026 Valencia, Spain; rubio_lui@gva.es (L.A.R.-M.); empar13@hotmail.com (E.M.-A.); 2Pathology Department, University of Valencia, 46010 Valencia, Spain; emilio.medinacs@gmail.com (E.M.-C.); isidro.machado@uv.es (I.M.); samuel.navarro@uv.es (S.N.); monicaem956@gmail.com (M.E.); antonio.llombart@uv.es (A.L.-B.); 3Molecular Biology Department, Instituto Valenciano de Oncología, 46009 Valencia, Spain; rlopez@fivo.org (R.L.-R.); jalopez@fivo.org (J.A.L.-G.); 4Pathology Department, Instituto Valenciano de Oncología and Patologika Laboratory Hospital Quiron Salud, 46010 Valencia, Spain; 5Department of Pathology, Catholic University of Valencia, 46001 Valencia, Spain; 6Joint Cancer Research Unit, Centro de Investigación Príncipe Felipe (CIPF), 46012 Valencia, Spain; 7Pathology Department, Hospital Clínic Universitari, 46010 Valencia, Spain; 8Instituto Salud Carlos III (CIBERONC), 28220 Madrid, Spain

**Keywords:** synovial sarcoma, BCOR, GLI1, immunohistochemistry, prognostic factors

## Abstract

Synovial sarcoma (SS) is a rare soft-tissue tumor characterized by a monomorphic blue spindle cell histology and variable epithelial differentiation. Morphologically, SSs may be confused with other sarcomas. Systemic treatment is more effective for patients with high-risk SSs, patients with advanced disease, and younger patients. However, further studies are required to find new prognostic biomarkers. Herein, we describe the morphological, molecular, and clinical findings, using a wide immunohistochemical panel, of a series of SS cases. We studied 52 cases confirmed as SSs by morphological diagnosis and/or molecular studies. Clinical data (gender, age, tumor size, tumor location, resection margins, adjuvant treatment, recurrences, metastasis, and survival) were also retrieved for each patient. All the available H&E slides were examined by four pathologists. Three tissue microarrays (TMAs) were constructed for each of the tumors, and a wide immunohistochemical panel was performed. For time-to-event variables, survival analysis was performed using Kaplan–Meier curves and log-rank testing, or Cox regression. Statistical significance was considered at *p* < 0.05. The mean age of our patients was 40.33, and the median was 40.5 years. We found a predominance of males versus females (1.7:1). The most frequent morphological subtype was monophasic. TRPS1, SS18-SSX, and SSX-C-terminus were positive in 96% of cases. GLI1 expression was strong in six and focal (cytoplasmic) in twenty patients. Moreover, BCOR was expressed in more than half of SSs. Positive expression of both proteins, BCOR and GLI1, was correlated with a worse prognosis. Multivariate analysis was also performed, but only BCOR expression appeared to be significant. The combination of GLI1 and BCOR antibodies can be used to group SSs into three risk groups (low, intermediate, and high risk). We hypothesize that these findings could identify which patients would benefit from receiving adjuvant treatment and which would not. Moreover, these markers could represent therapeutic targets in advanced stages. However, further, larger series of SSs and molecular studies are necessary to corroborate our present findings.

## 1. Introduction

Synovial sarcoma (SS) is a rare soft-tissue tumor characterized by a monomorphic blue spindle cell histology showing variable epithelial differentiation. This sarcoma is characterized by a specific t(X;18)(p11;q11) translocation generating the *SS18::SSX1/2/4* fusion gene [1]. Incidences of SS are equally distributed between both sexes and may occur at any age, although 77% of cases appear before the age of 50 years [2]. The nature of this tumor is still unclear, and although recent studies may support a neural origin through TRPA1 expression [3], it remains classified as a sarcoma of uncertain differentiation. Neoplastic cells are dependent on SS18-SSX expression to maintain their transformed phenotype [4]. The majority of cases are also positive for BCL2 and CD99, and less than half the neoplasms show expression of smooth muscle actin (SMA) and S100 [5]. However, all these markers are nonspecific. 

Glioma-associated oncogene 1 (GLI1) transcription factor is expressed as a result of Sonic hedgehog pathway activation, the dysregulation of which contributes to tumorigenesis in several tumor types [6,7]. Histologically, epithelioid SSs may be confused with GLI tumors, and the expression of GLI1 in SSs remains unclear as few studies have been performed with GLI1 antibodies [6,8]. BCL6 corepressor (BCOR) gene alteration is a genetic signature of rare subsets of sarcomas. BCOR is a highly sensitive marker in *BCOR* gene-rearranged tumors but is also often observed in SSs (around half the neoplasms) [9,10]. Moderate or strong nuclear staining for the transcriptional corepressor TLE1 is found in the large majority of SSs, but its staining is not specific to SSs as it may also be expressed in malignant peripheral nerve sheath tumors and solitary fibrous tumors [5,11,12]. Recently, two novel antibodies have been developed to detect the SS18-SSX fusion protein: an SS18-SSX fusion-specific antibody (E9X9V clone) that binds to amino acid residues surrounding the SS18-SSX fusion site [13,14] and an SSX-specific antibody (E5A2C clone) that binds to the C-terminus of the SSX protein [15]. These two antibodies were shown in subsequent studies to be good surrogates for FISH testing [16,17,18]. In more recent studies, the chimeric SS18-SSX fusion protein has been shown to represent the main driver of tumorigenesis, promoting CREB expression and phosphorylation and providing evidence for molecularly targeted therapies [19]. 

Trichorhinophalangeal syndrome 1 (TRPS1) protein is known to be a modulator in the mesenchymal-to-epithelial transition during the development of multiple tissue types, including cartilage, bones, kidneys, and hair follicles [20,21,22]. More recently, TRPS1 was discovered to belong to the GATA family of transcription factors and functions as a crucial regulator for the growth and differentiation of normal breast epithelial cells [23]. In recent publications, TRPS1 has been shown to be more expressed in SSs compared with other soft-tissue tumors, and its expression is related to the activity of the SS18-SSX fusion oncoprotein [24].

Despite all these recent advances, SSs continue to have a variable prognosis. Increasing size, age, and tumor grade have been demonstrated to be negative predictive factors for both local disease recurrence and metastasis. Metastatic disease commonly occurs in the lungs and bones but is also found in regional lymph nodes. Wide surgical excision remains the standard of care for definitive treatment, with adjuvant radiation used for larger and deeper neoplasms [1,25,26]. In general, systemic treatment is more effective for patients with high-risk SSs, patients with advanced disease, and younger patients [27], and some phase II clinical trials have combined gemcitabine and docetaxel in metastatic or unresectable locally advanced SSs [28]. The *BRAF* V600E mutation has been described in a small subset of SSs, offering a potential therapeutic target [29]. Further studies are required to find characteristic and prognostic factors. However, the rarity of this tumor makes it difficult to collect large series from one single institution.

Herein, we describe the morphological, molecular, and clinical findings, with a wide immunohistochemical panel, of a series of SSs collected in order to find new prognostic markers and define their biological correlations with respect to clinical outcome. 

## 2. Results

### 2.1. Clinical Findings

The clinical parameters are summarized in Table 1. The mean age of our patients was 40.33 years, with a range between 9 and 82 years and a median of 40.5 years. We found a predominance of males with 33 patients versus 19 females (1.7:1). The majority of SSs were located in the limbs (54%), with the thigh and knee being the main sites. The second most common location was the trunk (46%); three cases were reported in the upper limbs and one case in the head (orbit). In two cases, location data were not available. The majority of patients presented clinical stage II (56.10%); see Table 1. 

The mean tumor size was 9.11 cm, with the largest being 20 cm, reported in the right thigh. The median size was 9 cm (range 0.8–20 cm). Surgical margins were reported in 43 patients, of which 32.56% were positive and 60.46% were negative. Three cases were considered unresectable, and neoadjuvant therapy was administered to 23.26% of patients.

Local recurrence and metastasis were observed in 17 (32.69%) and 23 patients (44.23%), respectively. All distant metastases were to the lung. The mean PFS was 55.3 months, with a median of 33 months. The median OS was 50 months (range: 2–336 months). SS was the direct cause of death in 19 of the 42 deceased patients (45%).

### 2.2. Histopathological Findings

All SSs were classified histologically according to WHO criteria as monophasic, biphasic, and undifferentiated tumors. The most frequent morphological subtype was monophasic (65%), followed by biphasic (27%), and only four tumors were undifferentiated (8%) (Figure 1). 

### 2.3. Immunohistochemical Findings

The main immunohistochemical results are shown in Table 2. Results for the remaining analyzed proteins are described in Appendix A. In our study, almost all neoplasms were positive for BCL2, CD99, CD56, FLI1, TLE1, MUC4, and PDGFR alpha. Most tumors also expressed epithelial markers such as EMA and pancytokeratin. TRPS1 (Figure 2A), H3K27me3, SS18-SSX, and SSX-C-terminus were positive in most cases, while 20% of cases presented reduced INI1 expression. NKX2.2 expression was weak to intense in half the tumors (Figure 2B). BCOR was expressed in 56% of patients, and interestingly, GLI1 expression was nuclear in six cases and cytoplasmic in twenty cases (Figure 2C–F). However, GATA3 and Her2 were negative in all SSs. 

The proliferative index with Ki67 was high, intermediate, and low at 26%, 19%, and 55%, respectively.

### 2.4. Molecular Analysis

All SSs were studied by FISH as described in the Section 2. The *SYT* rearrangement was found in all 52 selected cases (Figure 3).

### 2.5. Statistical Study

The prognostic significance of IHC protein expression was evaluated with log-rank testing (Appendix A). In this regard, BCOR (*p* = 0.0024) and GLI1 (*p* = 0.0037) proteins had an impact on PFS (Figure 4). BCOR expression also showed a correlation with OS (*p* = 0.011). Positive expression of both proteins, BCOR (HR = 7.1, 1.92–26; *p* = 0.003) and GLI1 (HR = 8, 1.91–33.8; *p* = 0.004), was correlated with a worse prognosis. Multivariate analysis was also performed, but only BCOR appeared to be significant.

In order to establish the expression profile of the cases based on the patterns of both significant proteins, BCOR and GLI1, a categorization of cases was performed. Three risk groups resulted from the re-classification. The first (low-risk) group comprised the cases with negative expressions of BCOR and GLI1, or cases that mixed a focal expression of one and a negative of the other. The third (high-risk) group comprised the cases with positive expressions of both proteins, while the intermediate-risk group presented all the remaining combinations (Figure 5B). This classification increases the prognostic impact of single-protein analysis, obtaining a *p*-value of 0.00001 in the log-rank test (Figure 5A). In this case, not only positive cases presented an impact on relapse risk (HR = 15.7, 3.6–69; *p* > 0.001) but also intermediate cases (HR = 3.7, 1.1–12; *p* = 0.034) (Figure 5B).

## 3. Discussion

SS is an infrequent soft-tissue tumor characterized by a monomorphic spindle cell histology and variable epithelial differentiation. In recent publications, the TRPS1 protein has been shown to be expressed in SSs, being related to the activity of the SS18-SSX fusion oncoprotein [24]. The high expression of TRPS1 in our series confirms the results of Cloutier J.M. et al. [24], even though this antibody has also been described in other tumors such as breast carcinoma [30]. It suggests that this protein is a good marker for SSs, complementing the SS18-SSX fusion oncoprotein. The expression of different markers such as BCOR and NKX2.2 implies a diagnostic conflict and overlap with other similar small round cell entities such as BCOR-altered neoplasms and Ewing sarcomas. It is therefore important to use other complementary antibodies or molecular studies to determine an approach for each diagnosis.

NKX2.2 has been proven to be a very specific and sensitive marker for the diagnosis and detection of Ewing sarcoma, especially where the *EWSR1::FLI1* translocation exists [31]. However, it is well known that NKX2.2 is a sensitive but imperfectly specific marker for Ewing sarcoma [32]. In a more recent publication, Saeed SM et al. [33] studied this marker in other small round cell tumors. They found that 12% of SSs were positive for NKX2.2. Similar results were obtained by Hung Y.P. et al. [32], with expression in 10% of SSs. These results are lower than in our series, where NKX2.2 achieved a percentage of approximately 27% in SSs. Other studies have observed an amplification of the *MDM2* gene with a frequency as high as 40% [34], although this finding has not been confirmed by other subsequent studies [35,36,37]. MDM2 expression and its correlation with p53 expression have recently been corroborated by Larque et al. [38] in a wide series of SSs. We observed MDM2 expression in SSs in a similar proportion to other studies [34,38], although we could not find a correlation with p53 expression in our short series. 

The identification of *BCOR* gene alteration has recently contributed to the definition of new entities in the current WHO (2020) classification of soft-tissue and bone tumors [39]. BCOR is a highly sensitive marker for identifying small round cell sarcomas with *BCOR* gene alteration, although other reports have suggested that BCOR is less specific than CCNB3 in the diagnosis of *BCOR::CCNB3* sarcomas (BCSs) and is often observed in SS [9,10]. In our study, more than half of the tumors expressed partial and strong positivity for BCOR in a similar proportion to Kao Y.C. et al. [9]. This finding implies a differential diagnostic challenge between SSs and BCSs, more so given that BCSs are also usually positive for TLE1 and CD99. Interestingly, in our series, BCOR expression presented an impact on PFS and OS. As far as we know, BCOR expression has not been described as a significant prognostic factor in SSs so far. However, in a recent paper, the *BCOR* mutation has been correlated with a worse prognosis in hematologic neoplasms [40]. 

*GLI1* amplification and gene fusions have been identified in multiple mesenchymal neoplasms and an emerging class of *GLI1*-altered mesenchymal tumors [6,7]. In previous reports, Stein U. et al. [8] described the expression of the *GLI* gene in SSs. However, more recently, Parrack PH et al. [6] performed GLI1 immunohistochemistry on 10 biphasic SSs, finding no tumors positive for GLI1. Nevertheless, in our study, we observed cytoplasmic or focal GLI1 expression in 20 patients, although only six tumors presented nuclear and positive staining. Interestingly, of these six cases, two belonged to undifferentiated SSs and four to a monophasic subtype. No biphasic subtype presented nuclear or intense expression for GLI1 in our series. Similar to BCOR, GLI1 expression correlates with shorter disease-free periods. Recently, GLI1 was found to be significantly associated with a worse prognosis in oral, gastric, and prostatic cancer patients [41,42,43]. Our results with regard to SS show a correlation with these studies of other neoplasms. However, this finding has not yet been studied and observed in SSs and cannot be corroborated by other studies. We hypothesize that the combination of BCOR and GLI1 expression in SSs could predict biological behavior and may provide a possible opportunity as a therapeutic target in these high-grade sarcomas. Recently, Li et al. [44] associated the high expression of pAkt, pmTOR, and p4E-BP1 with aggressive clinical behavior in SSs and provided evidence for prognostic evaluation and targeted therapy. Other studies have provided further evidence that aberrant activation of the Wnt/β-catenin pathway is present in most SSs [45]. Studies of the prognostic significance of the *SS18::SSX1/2/4* fusion type have yielded inconsistent results, either positive for modest effects or negative [46,47].

Nevertheless, according to our results, GLI1, NKX2.2, and BCOR can also be expressed in SSs, representing a challenge in differential diagnosis with other small round/spindle cell sarcomas. Moreover, we found that combined GLI1 and BCOR expression can correlate with a worse prognosis for local recurrence and overall survival in SSs. Our study is limited by its retrospective nature and relatively small series. Further larger series and molecular studies, especially of *BCOR*, *GLI1*, and *MDM2* gene status, are necessary to corroborate our present findings.

## 4. Materials and Methods

### 4.1. Patients and Samples

Fifty-six cases of SSs diagnosed between 2006 and 2022 were collected from the pathology departments of the Hospital Universitari i Politècnic La Fe, Valencia, and the Hospital Clínic Universitari, Valencia. The final study comprised 52 cases confirmed as SSs by morphological diagnosis and/or molecular studies. Tissue samples were stained with hematoxylin and eosin (H&E) for histological analysis. Approval for this study was obtained from the Ethics Committee of the Universitat de València Estudi General (UVEG). Clinical data (gender, age, tumor size, tumor location, resection margins, adjuvant treatment, recurrences, metastasis, and survival) were also retrieved, as shown in Table 1. Complete clinical reports were collected in 44 out of the 52 cases. The histology was reviewed, and an immunohistochemical panel was performed in all 52 cases.

### 4.2. Histopathology

All the available H&E slides were examined by four pathologists (F.G., E.M.-C., I.M., and A.L.-B.), all blinded to the clinical data. In cases of disagreement, a consensus was reached on a multi-head microscope. SS diagnosis was established according to World Health Organization (WHO) criteria, classifying SSs as monophasic, biphasic, or poorly differentiated [1]. The most representative areas of each tumor were chosen for inclusion within tissue microarrays.

### 4.3. Assembly of Tissue Microarrays (TMAs) 

Three tissue microarrays for each tumor were performed using a manual tissue microarray instrument (Beecher Instruments, Sun Prairie, WI, USA). Three cores (1 mm in thickness) of each sample were included. 

Following TMA construction, a hematoxylin and eosin (H&E)-stained section of each TMA was performed to confirm the presence of an intact and representative neoplasm. In addition, 3 µm sections were cut in order to perform the immunohistochemical panel.

### 4.4. Immunohistochemistry

Immunohistochemistry was carried out on all TMA paraffin sections by an indirect peroxidase method as described in Appendix A. Antigen retrieval was performed with heat-induced epitope retrieval (autoclave at 1.5 atmospheres for 3 min in citrate buffer). Bound antibodies were visualized by an avidin–biotin–peroxidase procedure (LSAB Agilent^®^). Appropriate positive and negative controls were used for each antibody. Immunoreactivity was defined as follows: negative (0) when fewer than 5% of tumor cells were stained; focal or weak when 5–20%; and positive when more than 20% of tumor cells were stained. Immunoreactivity intensity was not categorized. For the proliferative index with Ki67, the scale was low (≤10%), moderate (11–20%), and high (>20%). All sections were independently evaluated by four pathologists (F.G., E.M.-C., I.M., and A.L.-B.), and in cases of disagreement, the score was determined by consensus. 

### 4.5. Fluorescence In Situ Hybridization (FISH)

Fluorescence in situ hybridization (FISH) was performed using the *SS18* Break-Apart Probe^®^ (Vysis, Abbott Laboratories, Hong Kong, China) to detect *SS18* (*SYT*) gene rearrangements on chromosome 18q11.2. All formalin-fixed, complete paraffin-embedded tissue sections (FPPEs) were pretreated, digested, and washed using the manual pretreatment Histology Accessory kit according to the manufacturer’s instructions (Dako, Glostrup, Denmark). For the *SS18* (*SYT*) gene evaluation, a minimum of 100 non-overlapping intact interphase nuclei sections were visualized at ×100 magnification with an oil immersion objective using an Axioscope 5 microscope with an Axiocam 305 mono camera and Colibri 5 LED illumination (ZEISS, Carl Zeiss Iberia, S.L., Jena, Germany). Finally, image processing and FISH picture acquisition were carried out with the ZEN 3.1 Blue Edition Imaging Software (ZEISS, Carl Zeiss Iberia, S.L.). A case was considered positive for *SYT* gene rearrangement when at least 25 of 100 counted tumor cells (25%) showed separation between the red and green signals.

### 4.6. Statistical Analysis 

A specific database was constructed and implemented to collect the main clinical, histopathological, and immunohistochemical data of the patients. The Chi–square test was used to compare categorical clinicopathological variables. For time-to-event variables, survival analysis was performed using Kaplan–Meier curves and log-rank testing, or Cox regression. Statistical significance was considered at *p* < 0.05. All tests were two-tailed. The time-to-event variables investigated were progression-free survival (PFS), defined as the time between diagnosis of the disease and relapse or progression, and overall survival (OS), defined as the time between diagnosis and death.

## Figures and Tables

**Figure 1 ijms-25-07615-f001:**
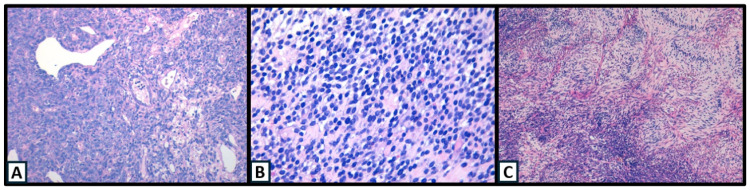
(**A**) H&E biphasic SS with glandular formation (40×); (**B**) H&E undifferentiated SS (40×); (**C**) H&E fusiform SS with Verocay Body-like structures (10×).

**Figure 2 ijms-25-07615-f002:**
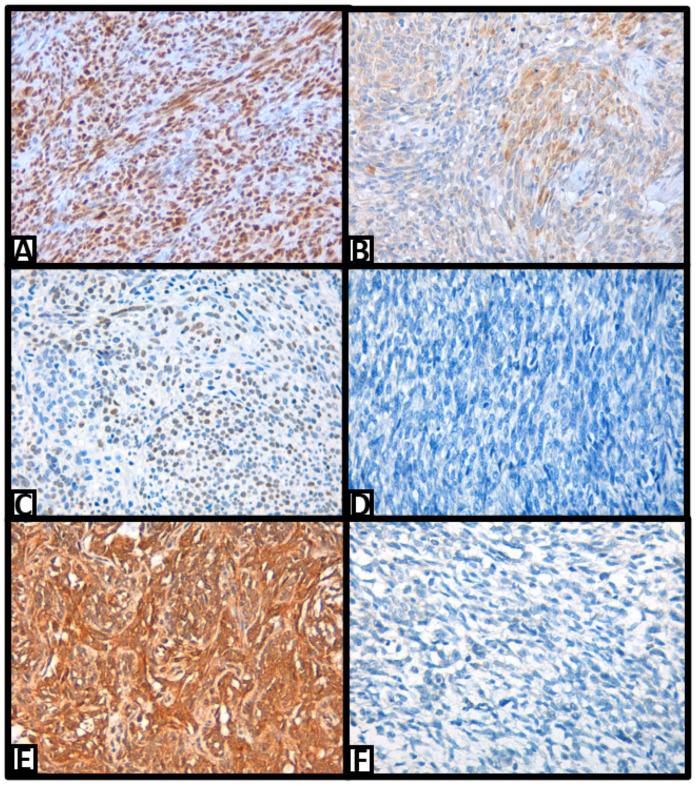
(**A**) TRPS1 nuclear expression (40×); (**B**) NKX2.2 nuclear expression (40×); (**C**) SS with BCOR nuclear expression (40×); (**D**) SS BCOR: negative (40×); (**E**) SS GLI1: positive (40×); (**F**) SS GLI1: negative (40×).

**Figure 3 ijms-25-07615-f003:**
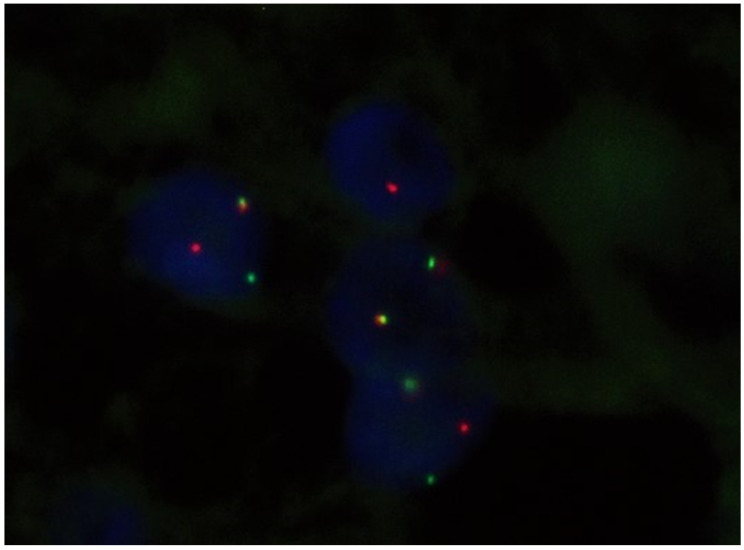
Rearrangement of the SYT gene (SS18) located in the 18q11.2 region using the FISH technique in a paraffinized tissue section. SYT gene (green signal).

**Figure 4 ijms-25-07615-f004:**
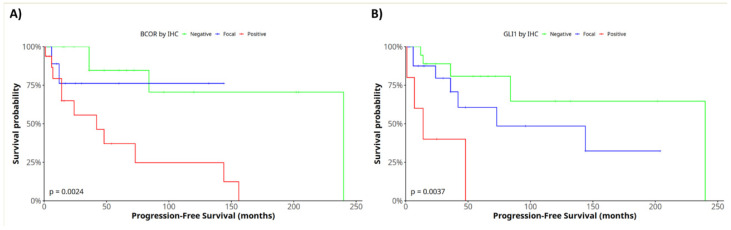
Log-rank test to evaluate the prognostic value of IHC-based protein expression with respect to PFS. Kaplan–Meier curves for (**A**) BCOR expression and (**B**) GLI1 expression.

**Figure 5 ijms-25-07615-f005:**
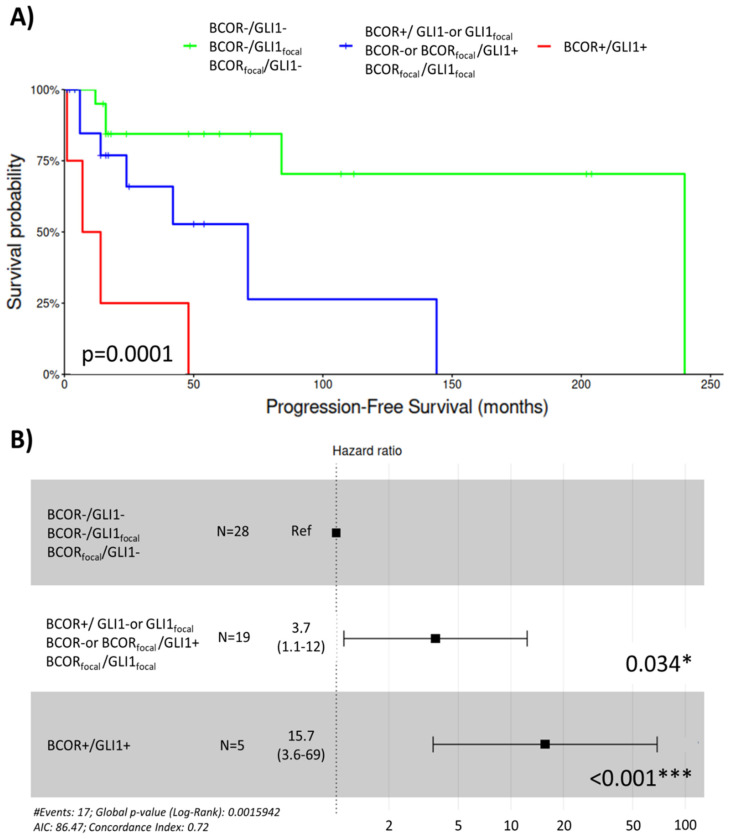
Prognostic value of integrated IHC-based BCOR and GLI1 expression profiles by (**A**) log-rank test and (**B**) Univariate Cox regression (* *p*-value ≤ 0.05, *** *p*-value ≤ 0.001).

**Table 1 ijms-25-07615-t001:** Demographic, morphological and clinical features of patients.

Parameter	Value
**Age (years) (*n* = 52)**	
Mean	40.33
Median (range)	40.5 (9–82)
**Sex (*n* = 52)**	
F	19 (36.54%)
M	33 (63.46%)
**Location (*n* = 50)**	
Limbs	27 (54.00%)
Trunk and head	23 (46.00%)
**Size (cm) (*n* = 22)**	
Mean	9.11
Median (range)	9 (0.8–20.00)
**PFS (months) (*n* = 44)**	
Mean	55.3
Median (range)	33 (1–240)
**OS (months) (*n* = 52)**	
Mean	79.55
Median (range)	50 (2–336)
**Stage (*n* = 41)**	
II	23 (56.10%)
III	10 (24.40%)
IV	8 (19.50%)
**Histology (*n* = 52)**	
Monophasic	32 (61.54%)
Biphasic	13 (25.00%)
Undifferentiated	7 (13.46%)
**Neoadjuvant therapy (*n* = 43)**	
No	33 (76.74%)
Yes	10 (23.26%)
**Surgical margins (*n* = 43)**	
Free	26 (60.46%)
Affected	14 (32.56%)
Unresectable	3 (6.98%)
**Local recurrence (*n* = 52)**	
No	35 (67.31%)
Yes	17 (32.69%)
**Metastasis (*n* = 52)**	
No	29 (55.77%)
Yes	23 (44.23%)
**Exitus by SS (*n* = 42)**	
No	23 (54.76%)
Yes	19 (45.24%)

PFS: Progression-free survival; OS: Overall survival; SS: Synovial Sarcoma.

**Table 2 ijms-25-07615-t002:** Results of some relevant immunohistochemical markers.

Antibody	Value
**Ki67 (*n* = 42)**	
0–10%	23 (55%)
11–20%	8 (19%)
>20%	11 (26%)
**BCOR (*n* = 52)**	
Negative	23 (44.23%)
Focal	11 (21.15%)
Positive	18 (34.62%)
**p53 (*n* = 50)**	
Negative	11 (22%)
Focal	15 (30%)
Positive	24 (48%)
**GLI1 (*n* = 51)**	
Negative	25 (49.02%)
Focal	20 (39.22%)
Positive	6 (11.76%)
**p16 (*n* = 51)**	
Negative	32 (62.75%)
Focal	11 (21.57%)
Positive	8 (15.69%)
**CD117 (*n* = 50)**	
Negative	29 (58%)
Focal	19 (38%)
Positive	2 (4%)

## Data Availability

The original contributions presented in the study are included in the article/Appendix A, further inquiries can be directed to the corresponding author.

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
