# Peer review of "The Combined Immunohistochemical Expression of GLI1 and BCOR in Synovial Sarcomas for the Identification of Three Risk Groups and Their Prognostic Outcomes: A Study of 52 Patients"

_ijms, 2024, doi:10.3390/ijms25147615_

Round 1
Reviewer 1 Report
Comments and Suggestions for Authors
For my opinion this paper is interesting and well conducted. Before acceptance of the manuscript, the authors need to address the following things:
Materials and methods:
Table suppl 1 shows MUC4 antibody which is then not discussed in the results.
Results:
Line-183 Figure 1 and legend are not clear:
-Why the authors insert (H&E) together with TRPS1 protein expression?
Line 187: To delete ??
Line 190: The authors describe INI1 in fig 1A but in the legend indicate the TRPS1 expression.
Table 2: The authors could also include the other results, if not in Table 2 in a supplementary table. They could also include some pictures.
FISH. Analysis: The authors could show a representative image
Fig.2. Why do the authors show both negative and positive BCOR expression and only positive for other proteins?
Fig.3. could be improved
Statistical analysis: the authors could show a supplemtary table with the results of the long rankig analysis for the most relevant proteins analyzed and the multivariate analysis
Author Response
Response to reviewers
Dear editor and reviewers,
We hereby resubmit our manuscript to be reconsidered for publication in International Journal of Molecular Sciences following a revision in accordance with the reviewers’ comments.
We would like to thank the reviewers for the evaluation of our manuscript and for their constructive suggestions which we believe have contributed towards improving the article.
With my best regards and thank you very much for evaluating our study.
Reviewer 1
For my opinion this paper is interesting and well conducted. Before acceptance of the manuscript, the authors need to address the following things:
Materials and methods:
- Table suppl 1 shows MUC4 antibody which is then not discussed in the results.
- We have added MUC4 results to Supplementary Table 2 and in the manuscript.
Results:
- Line-183 Figure 1 and legend are not clear: Why the authors insert (H&E) together with TRPS1 protein expression?
- We have moved TRPS1 picture to Figure 2A and we have added a H&E biphasic SS at Figure 1.
- Line 187: To delete ??
- ?? has been deleted
- Line 190: The authors describe INI1 in fig 1A but in the legend indicate the TRPS1 expression.
- We have changed the place of Fig. 1A to avoid misunderstanding.
- Table 2: The authors could also include the other results, if not in Table 2 in a supplementary table. They could also include some pictures.
- According to the reviewer’s suggestion, the information regarding the IHC results for the non-principally analyzed proteins has been added to a Supplementary Table 2. Reference to the Supplementary Table has also been included in the main manuscript.
- Newly added items:
- Supplementary Table 2 (Table S2): Immunohistochemistry results for complementary analyzed proteins.
- Line 212-213: “Results for remaining analyzed proteins are collected in Supplementary Table 2”
- Analysis: The authors could show a representative image
- We have added Figure 3 with FISH picture and we have renumbered Figures 3, 4 and 5.
- 2. Why do the authors show both negative and positive BCOR expression and only positive for other proteins?
- We have added a new negative picture of GLI1 in figure 2. In this way, the two most relevant markers are represented in the positive and negative form.
- 3. could be improved
- Following the aforementioned recommendations of the reviewer, quality and clarity of Figure 3 has been improved. Due to the lack of concrete modifications, in addition to global quality of the figure, axis and legends’ sizes have been enlarged, aiming to clarify the information contained.
- Statistical analysis: the authors could show a supplementary table with the results of the log rank analysis for the most relevant proteins analyzed and the multivariate analysis
- As pointed out by the reviewer, a supplementary table involving results for log-rank testing has been included. Reference to supplementary table has also been added in the main manuscript. Due to the absence of significant results for multivariate analysis, beyond BCOR expression, the inclusion of such information in the supplementary table has been considered non-informative.
- Newly added items:
- Supplementary Table 3 (Table S3): Results of Log-rank test to evaluate the value of main IHC-based protein expression data regarding Progression-Free Survival and Overall Survival.
- Line 229-230: “The prognostic significance of the level of protein expression, obtained by IHC, was evaluated with log-rank testing (Supplementary Table 3).”
Reviewer 2 Report
Comments and Suggestions for Authors
The authors investigate and claim that combined immunohistochemical expression of GLI1 and BCOR in synovial sarcoma identifies three risk groups and their prognostic outcome. A study of 52 patients. The study is so interesting, however I have some concerns to discuss.
-What statistical methods were used to obtain the results that the expression of GLI1 and BCOR had a negative prognostic impact?
-What multivariate analysis was used to obtain the result that the expression of BCOR was significant?
-You conclude that the combination of GLI1 and BCOR antibodies can divide SS into three risk groups (low, intermediate and high risk), but what specific criteria were used to divide these risk groups?
-How were the tissue microarrays (TMAs) used in this study prepared and what immunohistochemical panels were performed?
-You state that a larger series of SS and molecular studies are needed to corroborate the results of this study, what additional studies do you specifically consider necessary?
-Is the same true for neurogenic synovial sarcomas? The following literature.
Intraneural synovial sarcoma of the tibial nerve. Rare Tumors. 2018;10:2036361318776495. Published 2018 May 24. doi:10.1177/2036361318776495
Author Response
Dear editor and reviewers,
We hereby resubmit our manuscript to be reconsidered for publication in International Journal of Molecular Sciences following a revision in accordance with the reviewers’ comments.
We would like to thank the reviewers for the evaluation of our manuscript and for their constructive suggestions which we believe have contributed towards improving the article.
With my best regards and thank you very much for evaluating our study.
Reviewer 2
The authors investigate and claim that combined immunohistochemical expression of GLI1 and BCOR in synovial sarcoma identifies three risk groups and their prognostic outcome. A study of 52 patients. The study is so interesting, however I have some concerns to discuss.
- What statistical methods were used to obtain the results that the expression of GLI1 and BCOR had a negative prognostic impact?
- After stratification of patients based on IHC results for BCOR and GLI1 protein expression, log-rank test has been performed to establish the prognostic impact. Kaplan-Meier curves have also been obtained to visualize the results. In addition to log-rank testing and aiming to obtain the hazard ratio of the BCOR-GLI1 combined classification, an univariate cox regression test and forest plot were performed.
- This information is contained in the main manuscript in the following sections:
- Material and methods (Statistical analysis sub-section): line 180 “For time-to-event variables, survival analysis was performed using Kaplan–Meier curves, and log-rank testing or Cox regression. Statistical significance was considered at p < 0.05. All tests were two-tailed.”
- Results (Statistical study sub-section): line 229 “The prognostic significance of the level of protein expression, obtained by IHC, was evaluated with log-rank testing (Supplementary Table 3).”
- Figure 4 and 5 captions:
- 4: “Log-rank test to evaluate the prognostic value of IHC-based protein expression with respect to PFS. Kaplan-Meier curves for (A) BCOR expression (B) GLI1 expression.”
- 5: “Prognostic value of integrated IHC-based BCOR and GLI1 expression profiles by (A) Log-Rank test and (B) Univariate Cox regression.”
- What multivariate analysis was used to obtain the result that the expression of BCOR was significant?
- The multivariate test applied to decipher the independent prognostic value of the IHC-based protein expression is a Cox Regression analysis. This information is contained in the main manuscripts in the following section:
- Material and methods (Statistical analysis sub-section): line 180 “For time-to-event variables, survival analysis was performed using Kaplan–Meier curves, and log-rank testing or Cox regression. Statistical significance was considered at p < 0.05. All tests were two-tailed.”
- You conclude that the combination of GLI1 and BCOR antibodies can divide SS into three risk groups (low, intermediate and high risk), but what specific criteria were used to divide these risk groups?
- The stratification of patients regarding the combination of BCOR and GLI1 protein expression results was built based on the prognostic power. Different combinations were tested during the analytical steps, the selected combination was the most powerful regarding prognosis. In addition to the prognostic impact, the stratification was also biologically significant, representing, in a very simplified manner double-negative, double-positive and mixed groups.
- How were the tissue microarrays (TMAs) used in this study prepared and what immunohistochemical panels were performed?
- The TMA was assembled using a manual tissue arrayer (Beecher Instruments, Sun Prairie, WI), incorporating three cores each (1 mm in diameter) of each patient, according to manufacturer instructions.
- The immunohistochemical panel used is included in Supplementary Table 1.
- You state that a larger series of SS and molecular studies are needed to corroborate the results of this study, what additional studies do you specifically consider necessary?
- Given the need to validate the results shown in the manuscript, establishing the protein expression status using the same methodology in an independent real-world series of patients diagnosed with synovial sarcoma is highly recommended. Thus, the steps to follow would be; collect a prospective, independent and larger series, perform the immunohistochemical analysis, stratify patients based on the IHC results as presented in the manuscript, and determine the prognostic impact.
- Is the same true for neurogenic synovial sarcomas? The following literature.
Intraneural synovial sarcoma of the tibial nerve. Rare Tumors. 2018;10:2036361318776495. Published 2018 May 24. doi:10.1177/2036361318776495
- We think that to know if the same thing occurs, we should study and perform BCOR and GLI1 in this type of synovial sarcomas, gathering a large series. This type of SS has been described in isolated case reports but a serial study of these subtypes of SS has not been performed. In our series we did not have any neurogenic synovial sarcoma. We assume the same results should come out.
Round 2
Reviewer 2 Report
Comments and Suggestions for Authors
The manuscript is suitable for publication.